# Molecular Determinants of Calcitriol Signaling and Sensitivity in Glioma Stem-like Cells

**DOI:** 10.3390/cancers15215249

**Published:** 2023-10-31

**Authors:** Sarah Rehbein, Anna-Lena Possmayer, Süleyman Bozkurt, Catharina Lotsch, Julia Gerstmeier, Michael Burger, Stefan Momma, Claudia Maletzki, Carl Friedrich Classen, Thomas M. Freiman, Daniel Dubinski, Katrin Lamszus, Brett W. Stringer, Christel Herold-Mende, Christian Münch, Donat Kögel, Benedikt Linder

**Affiliations:** 1Experimental Neurosurgery, Department of Neurosurgery, Neuroscience Center, Goethe University Hospital, 60596 Frankfurt am Main, Germany; sarahrehbein@t-online.de (S.R.); anna-lenapossmayer@web.de (A.-L.P.); julia.gerstmeier@outlook.de (J.G.); koegel@em.uni-frankfurt.de (D.K.); 2Faculty of Medicine, Institute of Biochemistry II, Goethe University Frankfurt, 60596 Frankfurt am Main, Germany; bozkurt@med.uni-frankfurt.de (S.B.); ch.muench@em.uni-frankfurt.de (C.M.); 3Division of Experimental Neurosurgery, Department of Neurosurgery, University Hospital Heidelberg, INF400, 69120 Heidelberg, Germanychristel.herold-mende@med.uni-heidelberg.de (C.H.-M.); 4Dr. Senckenberg Institute of Neurooncology, Goethe University Hospital, 60596 Frankfurt am Main, Germany; michael.burger@kgu.de; 5Institute of Neurology (Edinger Institute), Frankfurt University Medical School, 60596 Frankfurt am Main, Germany; stefan.momma@kgu.de; 6Department of Medicine, Clinic III-Hematology, Oncology, Alliative Care Rostock, 18057 Rostock, Germany; claudia.maletzki@med.uni-rostock.de; 7Division of Pediatric Oncology, Hematology and Palliative Medicine Section, Department of Pediatrics and Adolescent Medicine, University Medicine Rostock, 18057 Rostock, Germany; carl-friedrich.classen@med.uni-rostock.de; 8Department of Neurosurgery, University Hospital Rostock, 18057 Rostock, Germany; thomas.freiman@med.uni-rostock.de (T.M.F.); daniel.dubinski@med.uni-rostock.de (D.D.); 9Department of Neurosurgery, University Medical Center Hamburg—Eppendorf, 20251 Hamburg, Germany; lamszus@uke.uni-hamburg.de; 10College of Medicine and Public Health, Flinders University, Sturt Rd., Bedford Park, SA 5042, Australia; brett.stringer@flinders.edu.au; 11German Cancer Consortium DKTK Partner Site Frankfurt/Main, 60590 Frankfurt am Main, Germany; 12German Cancer Research Center DKFZ, 69120 Heidelberg, Germany

**Keywords:** vitamin D, calcitriol, glioblastoma, glioma stem-like cells, cancer

## Abstract

**Simple Summary:**

Glioblastoma is one of the worst cancer types and the most common cancer originating within the brain. Patients afflicted by glioblastoma suffer from poor prognosis, a lack of specific therapies and frequent tumor recurrences. Many researchers are confident that glioblastoma cells can display traits of stem cells and that these attributes lead to an aggressive growth and high rate of recurrence. Based on our previous work that demonstrates that the “sun hormone” vitamin D_3_ can block these stem cell traits, we have now gained additional insights into the effects of the active form of vitamin D_3_, calcitriol. We can show that specific gene variants of the vitamin D_3_ receptor might be responsible for the sensitivity towards calcitriol and that sensitive cells are blocked in their stemness attributes as well as migratory potential.

**Abstract:**

Glioblastoma is the most common primary brain cancer in adults and represents one of the worst cancer diagnoses for patients. Suffering from a poor prognosis and limited treatment options, tumor recurrences are virtually inevitable. Additionally, treatment resistance is very common for this disease and worsens the prognosis. These and other factors are hypothesized to be largely due to the fact that glioblastoma cells are known to be able to obtain stem-like traits, thereby driving these phenotypes. Recently, we have shown that the in vitro and ex vivo treatment of glioblastoma stem-like cells with the hormonally active form of vitamin D_3_, calcitriol (1α,25(OH)_2_-vitamin D_3_) can block stemness in a subset of cell lines and reduce tumor growth. Here, we expanded our cell panel to over 40 different cultures and can show that, while half of the tested cell lines are sensitive, a quarter can be classified as high responders. Using genetic and proteomic analysis, we further determined that treatment success can be partially explained by specific polymorphism of the vitamin D_3_ receptor and that high responders display a proteome suggestive of blockade of stemness, as well as migratory potential.

## 1. Introduction

### 1.1. Glioblastoma

Glioblastoma is the most aggressive and most common primary malignant brain tumor among adults and is classified as a grade 4 tumor [1]. Despite decades of research, the prognosis is still very poor, and 5-year survival rates remain low [2]. Hence, novel approaches are desperately needed to address this issue. Research over the last few years has shown that one key determinant leading to treatment failure and tumor recurrences is tumor cells that can obtain a phenotype reminiscent of stem cells, so-called glioma stem-like cells (GSCs). These GSCs often express markers associated with undifferentiated cells, particularly of neurodevelopmental pathways, and are hypothesized to be able to replenish the tumor after treatment. Frequently employed marker proteins are the homeobox transcription factors SOX2 and SOX9, as well as the intermediate filament Nestin, while in a laboratory setting the ability to grow as spheres is considered a hallmark of stemness [3,4]. These marker proteins are essential for transforming GBM cells into GSCs and sustaining their GSC characteristics, enabling these cells to exhibit exceptional self-renewal abilities. Additionally, GSCs display increased mobility and invasiveness. The unique characteristics of GSCs, including their heterogenic phenotypes influenced by their microenvironment within GBM [5], pose significant challenges for targeted therapies. Consequently, these distinct features are the primary reasons why common treatment methods such as surgical tumor removal, adjuvant chemotherapy, or chemoradiotherapy often have a limited impact on GSCs. The primary cause of current therapeutic approaches’ ineffectiveness in treating GBM is the inability to eliminate GSCs within the tumor. Even when GSCs are in a dormant state, they still maintain their capacity for self-renewal. This means that only a tiny subset of GSCs that remains in the brain after treatment is enough to regenerate the entire tumor [6]. Therefore, to achieve a permanent and recurrence-free eradication of GBM, it is crucial to explore direct targeting of GSCs in addition to the existing treatment options. This approach holds great promise for GBM treatment. The current standard therapy for GBM is maximally possible surgical resection followed by radiochemotherapy using the alkylating agent temozolomide (TMZ), sometimes with additional treatment using tumor treating fields (TTF) [7,8,9,10]. Recently, we have shown that the hormonally active form of vitamin D_3_ (VitD_3_), calcitriol (1α,25(OH)_2_-vitamin D_3_), can block stemness programs in a subset of GSCs in vitro, leading to reduced tumor growth ex vivo [11].

### 1.2. Vitamin D_3_/Calcitriol and Its Antitumor Properties

The so-called “sun hormone” VitD_3_ and its natural or synthetic derivatives have been tested for their anticancer activity in a plethora of human cancers with mixed results, likely due to lack of proper stratification [12,13,14]. Physiologically, VitD_3_ is synthesized in the skin upon UV exposure and is metabolized into its active form by two hydroxylation steps in the kidney and liver and mainly acts via its cognate receptor, vitamin D receptor (VDR). Aside from this “classical” synthesis pathway, recent research has shown that most cells in the body express the necessary enzymes for local synthesis as well as the VDR [15,16]. Calcitriol binding to the VDR leads to dimerization of the VDR with the retinoid x receptor (RXR) and translocation of this heterodimer to the nucleus, where it regulates target gene expression [16,17]. In vitro studies have already indicated that calcitriol may offer a promising new therapeutic avenue for various cancer types. Furthermore, in vivo experiments have shown inhibition of tumor development in different tumor types [18,19,20]. Recent research suggests that both the levels of the progenitor of calcitriol (25(OH)-vitamin D_3_; calcidiol), as well as VDR expression, correlate with tumor burden and/or progression of brain tumors [21,22,23]. Our work could show that the sensitivity of GSCs strongly correlates with VDR expression and that the VDR expression can vary over at least three orders of magnitudes in vitro, suggesting a strong level of heterogeneity [11]. Additionally, we could show that calcitriol treatment of high-responding GSCs leads to a phenotype suggestive of stemness blockade. This resulted in an effective blockade of tumor growth using ex vivo murine, adult organotypic brain slice cultures transplanted with human GSCs. Additionally, our initial, albeit restricted dataset suggested that calcitriol evokes combined therapeutic effects with TMZ particularly in TMZ-resistant (MGMT-expressing GSCs with an unmethylated MGMT promoter) GSCs [11]. MGMT (O6-methylguanine-DNA methyltransferase) status is assessed clinically as a predictive marker for TMZ chemotherapy to guide therapy decisions as the MGMT enzyme can revert TMZ-induced DNA lesions. Hence, patients with MGMT expression are very resistant to TMZ and show little benefit from this aggressive treatment [24]. Hence, our data supports the proposition to employ calcitriol/VitD_3_ and/or their derivatives for adjuvant brain cancer treatment [25,26].

Here, we aimed to expand our cell panel and perform unbiased proteomic analysis of high- and non-responding GSCs to delineate differences in their response rates, and to identify key pathways involved in the response to calcitriol. To further support its potential clinical application, we performed additional ex vivo tumor growth assays, as well as treatment of freshly prepared, low-passage, patient-derived organoids using the combination of calcitriol and TMZ.

## 2. Materials and Methods

### 2.1. Cells and Cell Culture

For the experiments, 31 GSC lines from various sources were employed. The GBM patient-derived Q-Cell cell line resource consisting of BAH1, HW1, JK2, PB1, RKI1, RN1, SB2b and WK1 [27] were gifted from Bryan Day (Sid Faithful Brain Cancer Laboratory, QIMR Berghofer, Brisbane, Australia) and Brett Stringer (Flinders University, Adelaide, Australia). Q-Cell cell line characterization data are publicly available from Q-Cell https://www.qimrberghofer.edu.au/q-cell/ (last accessed on: 30 July 2023) [27,28,29]. NCH465, NCH601, NCH660h and NCH663 [30,31,32,33] were provided by Christel Herold-Mende (Division of Experimental Neurosurgery, Department of Neurosurgery, University Hospital Heidelberg, Heidelberg, Germany). Katrin Lamszus (Department of Neurosurgery, University Medical Center Hamburg-Eppendorf, Hamburg, Germany) provided GS-8, GS-73, GS-74, GS-80, GS-86, GS-90 and GS-101 [34,35]. HROG05, HROG52 and HROG63 [36] were provided by Carl-Friedrich Classen (Division of Pediatric Oncology, Hematology and Palliative Medicine Section, Department of Pediatrics and Adolescent Medicine, University Medicine Rostock, Rostock, Germany) and Claudia Maletzki (Department of Medicine, Clinic III-Hematology/Oncology/Palliative Care Rostock, Rostock, Germany). R28 was a kind gift from Dr. Christoph Beier (University Hospital Regensburg, Regensburg, Germany) and has been described [37,38]. GBM10, GBM15, MNOF35, MNOF69, MNOF76, MNOF107 and Beta4 [39,40] have been established at the Frankfurt site and we established an additional primary culture for this study: GBM21_01. The cell line was generated as described previously [41] and is derived from a 59-year-old, male patient after obtaining informed consent and ethics approval (Ethics Committee at the University Hospital Frankfurt; reference numbers SNO_NP_01-08, SNO-12-2016). The tumor was graded as IDH-WT WHO grade 4 with an unmethylated MGMT promoter.

The cell lines were cultured using the following media, based on the information from the providing groups: neurobasal medium (Gibco, Darmstadt, Germany) supplemented with 1 × B27; 100 U/mL Penicillin; 100 µg/mL streptomycin (P/S, Gibco); 1× GlutaMAX (Gibco); 20 ng/mL epidermal growth factor (EGF, Peprotech, Hamburg, Germany); and 20 ng/mL fibroblast growth factor (FGF, Peprotech). DMEM/F12 medium (Gibco, Darmstadt, Germany) containing 1× GlutaMAX, 20 ng/mL each of EGF and FGF, 20% BIT admixture supplement (Pelo Biotech, Planegg/Martinsried, Germany) and P/S. DMEM/F12 medium containing 1× GlutaMAX, 20 ng/mL each of EGF and FGF, 1 × B27 Supplement (Gibco) and P/S.

GFP-positive GSCs were generated as described previously [11] and below and supplemented with empirically determined concentrations of blasticidin (Sigma-Aldrich, Taufkirchen, Germany). Specifically, Beta4, GBM10, GS-101 and NCH481 received 3, 4.5, 8 and 2 µg/mL of blasticidin, respectively.

For comparative proteomics, all cell lines were cultured in the same medium to avoid medium-dependent artifacts. The following medium was used for this approach: DMEM/F12 medium containing 1× GlutaMAX, 20 ng/mL EGF and FGF, 1× B27 supplement and P/S.

HEK293T (ATCC #CRL-3216) were cultured in Dulbecco’s modified Eagle’s medium (DMEM GlutaMAX) supplied with heat-inactivated 10% FBS, 100 U/mL Penicillin and 100 µg/mL streptomycin (all from Gibco).

### 2.2. Limiting Dilution Assay

The limiting dilution assays were performed and analyzed as described previously [11]. Briefly, a descending 1:1 dilution starting from the first row was plated as freshly dissociated cell suspension in 96-well plates and treated with 50 nM calcitriol (Cayman Chemical, Ann Arbor, MI, USA) or solvent (EtOH, Sigma-Aldrich). After 7 days, the wells were checked for the presence of at least one sphere >50 µm and marked as positive. The ratio of positive and negative wells was entered into ELDA software using the standard-setting (http://bioinf.wehi.edu.au/software/elda; [42]; last accessed on 18 March 2023) and the stem-cell frequency was calculated. For figure generation, the data from at least two experiments were pooled and the fold change of the stem-cell frequency between solvent- and calcitriol-treated GSCs was calculated.

### 2.3. Taqman-Based qRT-PCR

In principle, the Taqman-based qRT-PCR was performed as described [11]. Briefly, 300,000 GSCs were manually dissociated via repetitive pipetting and cultured for 24 h prior to harvesting. RNA was isolated using the ExtractMe Total RNA Kit (Blirt S.A., Gdansk, Poland), while 1–2 µg RNA was used for cDNA synthesis. SuperScript III (Life Technologies, Darmstadt, Germany) was used for cDNA synthesis using 100 U of SSIII according to the manufacturer’s instructions. qRT-PCR was performed on a StepOne Plus System (Applied Biosystems, Darmstadt, Germany) in a 20 µL reaction volume using Taqman probes (Applied Biosystems, Darmstadt, Germany) and Fast-start Universal Probe Master Mix (Roche, Mannheim, Germany). Ct values were normalized to TATA box-binding protein (TBP). VDR expression was determined using the 2^−ΔΔCt^ method. The following Taqman probes were used: TBP (Hs00427620_m1) and VDR (Hs00172113_m1).

### 2.4. Lentiviral Transduction

GFP-positive GSCs were generated by transfecting 2 µg pLV[Exp]-EGFP.T2A.Bsd-CMV-ORF_Stuffer (VB900122-2891VDE; Vectorbuilder GmbH, Neu-Isenburg, Germany), 1.5 µg gag/pol plasmid (psPAX2, Addgene #12260) and 0.5 µg VSV-G envelope plasmid (pMD2.G, Addgene #12259) into HEK293T cells in 57 µL Opti-MEM and 6 µL FuGENE HD (Promega, Fitchburg, WI, USA) transfection reagent as described [11]. The viral supernatant was collected 16 and 40 h post transfection, pooled and applied to the GSCs after diluting 1:1 with fresh medium and the addition of protamine sulfate (Sigma-Aldrich) at a final concentration of 8 µg/mL. For transduction, 30,000 accutase-dissociated GSCs were seeded, transduced for 48 h and selected with the appropriate puromycin concentration. psPAX2 was a gift from Didier Trono (Addgene plasmid # 12260; http://n2t.net/addgene:12260; RRID:Addgene_12260; accessed on 22 March 2021). pMD2.G was a gift from Didier Trono (Addgene plasmid # 12259; http://n2t.net/addgene:12259; RRID:Addgene_12259; accessed on 22 March 2021).

### 2.5. Adult Organotypic Brain Slice Cultures and Ex Vivo Tumor Growth Assay

Adult organotypic brain slice cultures from murine brains (OTC) were performed as described previously [11,41,43]. Briefly, the brains were removed and sectioned into 150 µm transverse slices after embedding in 2% low-melting agarose using a vibratome VT1000 (Leica, Wetzlar, Germany). The OTCs were cultured on Millicell cell culture inserts in 6-well plates using 1 mL of FCS-free medium consisting of DMEM/F12 supplied with 1 × B27, 1 × N2 supplement and P/S (all from Gibco). Adequate spheres were generated by seeding manually dissociated GSCs 1–3 days prior in u-shaped 96-well plates and placed by hand using a P2.5 pipette with a minimal volume (~0.7 µL). One day after sphere spotting, pictures were taken (d0) using a Nikon SMZ25 stereomicroscope and treatment was started as indicated. Throughout the experiment, the treatment was renewed thrice a week using 1 mL of medium and tumor growth was monitored in regular intervals. The tumors were treated with 100 nM calcitriol or 50 µM TMZ, a combination of both, or solvent. Tumor growth was evaluated using FIJI (v1.52p) [44] and, for graphical presentation, the fold change of each tumor was calculated by normalizing to the d0 tumor sizes.

### 2.6. GBM Organoids Treatment with Calcitriol and/or Temozolomide

#### 2.6.1. Patient Samples

Human primary glioblastoma (GBM) tumor tissue samples (n = 9) were obtained from patients, who underwent surgical resection at the Department of Neurosurgery (University Hospital Heidelberg, Heidelberg, Germany). The usage of patient material was approved by the Institutional Review Board at the Medical Faculty Heidelberg (referral number: 005/2003). Informed consent was obtained from all patients included in the study. Patient characteristics are presented in Table 1.

#### 2.6.2. Generation of GBM Tumor Organoids

GBM-derived single-cell suspensions were used to generate tumor organoids by bioprinting cells in anti-adhesive 96-well plates at a density of 50,000 cells per well in a 150 μL culture medium. Tumor organoids were allowed to aggregate for 4 days.

#### 2.6.3. Live/Dead Staining of GBM Tumor Organoids

Cell viability in untreated tumor organoids was analyzed on day 7 using the LIVE/DEAD^TM^ Cell Imaging Kit (Invitrogen, Karlsruhe, Germany) according to the manufacturer’s instructions for exemplary cases.

#### 2.6.4. Treatment of GBM Tumor Organoids

Tumor organoids were treated on day 4, day 6 and day 8 with calcitriol (500 nM and 1000 nM) and/or Temozolomide (50 μM) while 0.1% EtOH and/or 0.5% DMSO served as the respective controls. On day 9, cell viability was analyzed using CellTiter-Glo^®^ 3D (Promega, Madison, WI, USA) assay according to the manufacturer’s instructions, and a Tecan Infinite^®^ 200 plate reader (Tecan Group Ltd., Männedorf, Switzerland) was used for recording luminescence measurement. Data were analyzed using GraphPad Prism 8.4.3 (GraphPad Software, Boston, MA, USA). For data analysis, a paired Student’s *t*-test was used. A *p* value * *p* < 0.05; ** *p* < 0.01 was considered to be significant. Data are presented as bar graphs showing the mean with standard error of the mean (SEM).

### 2.7. Restriction Fragment Length Polymorphism Analysis

Restriction Fragment Length Polymorphism (RFLP) analysis aims to identify DNA sequence polymorphisms in genes or DNA regions. DNA samples in question were digested with specific restriction endonucleases. Polymorphism can be identified as fragments of different lengths after restriction digestion [45]. Here, this technique was used to investigate if VDR polymorphisms play a role in the differential responses of high- and non-responding GCSs towards calcitriol.

Four VDR polymorphisms were considered: FokI (rs2228570) with alleles F and f with the genotype ff (all VDR DNA is restricted by FokI), indicating higher sensitivity of cells toward calcitriol, and the BsmI-ApaI-TaqI polymorphism (rs1544410, rs7975232, rs731236), with the “baT” genotype (BsmI and ApaI can digest the VDR DNA, TaqI cannot) leading to increased sensitivity [46].

#### 2.7.1. DNA-Extraction and PCR

To investigate which VDR polymorphisms are present in high- and non-responding GSCs, 100,000 cells of each high-responding and non-responding GSC line (Section 2.4) were transferred into 1.5 mL tubes and pelleted. DNA was extracted using 50–150 µL QuickExtract DNA Extraction Solution (LGC, Biosearch Technologies, Hoddesdon, UK), depending on the pellet size. Samples were heated to 65 °C for 6 min while shaking at 1000 rpm. Afterwards, samples were vortexed for 15 s before heating to 98 °C for 2 min. Extracted DNA was vortexed again and used directly or stored at −20 °C.

PCR primers were designed for specific amplification of the VDR fragments harboring the restriction sites for determination of the different polymorphism—FokI, BsmI, TaqI and ApaI—and specificity was checked using the web tool ‘Primer Blast’ on NCBI (https://www.ncbi.nlm.nih.gov/tools/primer-blast/; last accessed on 23 August 2020).

The following primers were used: VDR.FokI_f: 3′-AGCTGGCCCTGGCACTGACTCTGCTCT-5′; VDR.FokI_r: 3′-ATGGAAACACCTTGCTTCTTCTCCCTC-5′; VDR.BsmI_f: 3′-CAACCAAGACTACAAGTACCGCGTCAGTGA-5′; VDR.BsmI_r: 3′-AACCAGCGGAAGAGGTCAAGGG-5′; VDR.ApaI.TaqI_f: 3′-CAGAGCATGGACAGGGAGCAAG-5′; and VDR.ApaI.TaqI_r: 3′-GCAACTCCTCATGGCTGAGGTCTCA-5′. The PCR product obtained from the ApaI.TaqI pair was used to detect both polymorphisms.

For amplification via PCR, a master mix was prepared containing the following ingredients per DNA sample: 1 µL template DNA, 5 µL 10× Standard Taq Reaction Buffer (New England Biolabs, Frankfurt am Main, Germany) and 0.25 µL Taq-DNA polymerase (1.25 units/50 µL reaction), 200 mM dNTP mix, 0.5 µM each of both respective primers and for the BsmI and ApaI PCRs, 1.6 µL DMSO were added and filled to a final volume of 50 µL using PCR clean water.

PCR was performed using a thermocycler (Mastercycler, Eppendorf, Hamburg, Germany). Table 2 shows the parameters used for amplification of the different VDR polymorphisms:

#### 2.7.2. Restriction Digestion and Agarose Gel Electrophoresis

For restriction digestion of PCR amplified samples, DNA concentration was measured in each sample, and 1 µg DNA was mixed with 5 µL 10× rCutSmart Buffer and 0.25 µL of the respective restriction enzyme (TaqI-v2, FokI, ApaI or BsmI). The reaction mixture was filled up to 50 µL using H_2_O and incubated for 2 h to ensure complete DNA digestion. Incubation occurred at the respective optimal temperature of each enzyme as specified by the manufacturer. Lengths of PCR amplified polymorphisms as well as DNA fragments resulting from restriction digestion with the different restriction enzymes are shown in Table 3.

After digestion, DNA fragments were separated on a 2% agarose gel supplemented with Ethidium bromide (EtBr). Therefore, a 20 µL sample was mixed with 5 µL Gel Loading Dye. Also, 1 µg undigested sample was diluted with 19 µL H_2_O and mixed with 5 µL Gel Loading Dye. Digested as well as undigested samples that served as controls were subsequently loaded onto the gel wells. Additionally, 3 µL of a 100 bp marker was added into the outer wells of each line. The gel was run in Rotiphorese TAE buffer (Roth) at 100–130 V, depending on the gel size for 1 h. Pictures were taken under UV light using the Gel Documentation System E.A.S.Y Doc plus (Herolab) and analyzed via the software E.A.S.Y Win (v5.17.295).

### 2.8. Proteomics

#### 2.8.1. Sample Preparation for Mass Spectrometry

GSC spheroids were collected by centrifugation and washed with PBS prior to sample preparation. Further sample preparation was performed as described previously [11]. The cells were lysed and denatured with 2% SDS, 50 mM Tris-HCl pH 8, 150 mM NaCl, 10 mM TCEP, 40 mM chloroacetamide and protease inhibitor cocktail tablet (EDTA free, Roche) at 95 °C for 10 min, then sonicated for 1 min (1s ON/1s OFF pulse, 45% amplitude) and boiled again for 5 min at 95 °C. Pure proteins were obtained with methanol/chloroform precipitation. Protein pellets were then resuspended in Urea; 100 μg of protein per sample were digested overnight at 37 °C with LysC (Fujifilm Wako Chemicals Europe, Neuss, Germany) at 1:50 (*w*/*w*) ratio and Trypsin (Promega, V5113) at 1:100 (*w*/*w*) ratio. Digested peptides are purified using Sep-Pak tC18 cartridges (Waters, WAT054955). Peptide concentrations were determined with a μBCA assay (23235, Thermo Fisher Scientific, Langenselbold, Germany) and 10 μg of peptide per sample was labeled with TMTpro reagents (A44520, Thermo Fisher Scientific, Langenselbold, Germany). TMT-labeled samples were adjusted to equal amounts and pooled. Then, the pool was fractionated into 24 fractions using the high pH micro-flow fractionation method.

#### 2.8.2. High pH Micro-Flow Fractionation

Peptides were fractionated using high-pH liquid chromatography on a micro-flow HPLC (Dionex U3000 RSLC, Thermo Scientific). An amount of 45 µg of pooled and purified TMT-labeled peptides resuspended in Solvent A (5 mM ammonium-bicarbonate, 5%ACN) were separated on a C18 column (XSelect CSH, 1 mm × 150 mm, 3.5 µm particle size; waters) using a multistep gradient from 3–60% Solvent B (5 mM ammonium-bicarbonate, 90% ACN) over 65 min at a flow rate of 30 µL/min. Eluting peptides were collected every 43 s from minute 2 for 69 min into a total of 96 fractions, which were cross-concentrated into 24 fractions. Pooled fractions were dried in a vacuum concentrator and resuspended in 2% ACN, and 0.1% TFA for LC-MS analysis.

#### 2.8.3. Mass Spectrometry (LC-MS^3^)

Fractions were resuspended in 2% acetonitrile and 0.1% formic acid and separated on an Easy nLC 1200 (Thermo Fisher Scientific) and a 22 cm long, 75 μm ID fused-silica column, which had been packed in house with 1.9 μm C18 particles (ReproSil-Pur, Dr. Maisch, Ammerbuch-Entringen, Germany) and kept at 50 °C using an integrated column oven (Sonation). HPLC solvents consisted of 0.1% formic acid in water (Buffer A) and 0.1% formic acid and 80% acetonitrile in water (Buffer B). Assuming equal amounts in each fraction, 500 ng of peptides were eluted by a non-linear gradient from 7 to 40% B over 90 min, followed by a step-wise increase to 75% ACN in 6 min which was held for another 9 min. After that, peptides were directly sprayed into an Orbitrap Fusion Lumos mass spectrometer equipped with a nanoFlex ion source (Thermo Fisher Scientific).

Sprayed peptides were analyzed with the Multi-notch MS^3^-based TMT method in order to minimize ratio compression and ion interference as previously described [47] for total proteomics. Full scan MS spectra (350–1400 *m*/*z*) were acquired with a resolution of 120,000 at *m*/*z* 200, maximum injection time of 100 ms and AGC target value of 4 × 105. The most intense precursors with a charge state between 2 and 6 per full scan were selected for fragmentation (“Top Speed” with a cycle time of 1.5 s) and isolated with a quadrupole isolation window of 0.7 Th. MS2 scans were performed in the ion trap (Turbo) using a maximum injection time of 50 ms, AGC target value of 1.5 × 104 and fragmented using CID with a normalized collision energy (NCE) of 35%. SPS-MS3 scans for quantification were performed on the 10 most intense MS2 fragment ions with an isolation window of 0.7 Th (MS) and 2 *m*/*z* (MS2). Ions were fragmented using HCD with an NCE of 50% and analyzed in the Orbitrap with a resolution of 50,000 at *m*/*z* 200, a scan range of 100–500 *m*/*z*, AGC target value of 1.5 × 105 and a maximum injection time of 86 ms. Repeated sequencing of already acquired precursors was limited by setting a dynamic exclusion of 45 s and 7 ppm and advanced peak determination was deactivated.

#### 2.8.4. Proteomics Data Analysis

Raw data was analyzed with a Proteome Discoverer 2.4 (Thermo Fisher Scientific). SequenceHT node was selected for database searches of MS2 spectra. Human trypsin-digested proteome (Homo sapiens SwissProt database (TaxID:9606, version 12 March 2020)) was used for protein identifications. Contaminants (MaxQuant “contamination.fasta”) were determined for quality control. TMTpro (+304.207) at the N-terminus, TMTpro (K, +304.207) at lysine and carbamidomethyl (C, +57.021) at cysteine residues were set as fixed modifications. Methionine oxidation (M, +15.995) and acetylation (+42.011) at the protein N-terminus were set for dynamic modifications. Precursor mass tolerance was set to 7 ppm and fragment mass tolerance was set to 0.5 Da. Default percolator settings in PD were used to filter perfect spectrum matches (PSMs). Reporter ion quantification was achieved with default settings in consensus workflow. Protein file from PD was then exported to Excel for further processing. Normalized abundances from the protein file were used for statistical analysis after contaminations and complete empty values were removed. Significantly altered proteins were determined by two-sided, unpaired Student’s *t*-tests (*p*-value < 0.05), adding minimum log_2_ fold-change cut-off (≥0.5) with R version 4.0.2 in RStudio [48,49]. Correlation matrix analysis was performed using Instant Clue software (http://www.instantclue.uni-koeln.de/; last accessed on 2 March 2023) [50]. GraphPad Prism 9 was used for the volcano plot by plotting the log2 fold changes versus the–log10 *p*-value. Correlation plots were later edited with Adobe Illustrator CS5 (Adobe Inc., San Jose, CA, USA). For pathway-based analysis, we employed the STRING [51,52] (https://string-db.org/) and PANTHER [53,54,55,56] (http://www.pantherdb.org/) web apps. First, we submitted the significantly regulated proteins with an LFC >/< 0.5/−0.5 via the STRING portal to analyze which pathways are enriched among these groups. Secondly, we submitted our complete dataset consisting of gene name and the LFC value to perform an enrichment analysis using the standard settings. Lastly, we employed the PANTHER web app (PANTHER version 17.0, released 22 February 2022) to perform a statistical enrichment test (released 12 July 2022) by submitting again the gene name with the respective LFC value, selecting homo sapiens as background and FDR correction. The mass spectrometry proteomics data have been deposited to the ProteomeXchange Consortium via the PRIDE [57] partner repository with the dataset identifier PXD041634.

### 2.9. Statistics

Statistical analyses involved one-way and two-way ANOVA using GraphPad Prism 9 (GraphPad Software, La Jolla, CA, USA), with the respective post hoc test, as indicated. For LDA, the statistical evaluation was taken from ELDA software, which calculated statistical significance based on a chi2-square test (http://bioinf.wehi.edu.au/software/elda; [42]; last accessed on 18 March 2023).

## 3. Results

### 3.1. Calcitriol Reduces Sphere Formation of GSCs

Recently, we determined the antitumor activity of the hormonally active form of the “sun hormone” vitamin D_3_, calcitriol (1α,25(OH)_2_-D_3_), using a panel of 10 GSCs and were able to show that 6 of those displayed significantly reduced sphere formation following treatment, while 2 of these lines showed a particularly strong response [11]. To answer the questions of whether these effects are generalizable and if potential predictors for sensitivity (or lack thereof) can be found, we expanded our GSC panel by an additional 31 lines. This approach (Figure 1A) revealed that of all 41 analyzed GSC lines ~50 percent show a significant reduction of sphere-forming potential after treatment with 50 nM calcitriol (responders). By defining GSCs with at least a halving of the sphere formation potential as high responders, we determined that 50% of the responders, i.e., 10 GSC lines in total, can be classified accordingly. Next, we wondered, similar to our previous approach [11], whether calcitriol-sensitivity correlates with VDR expression and measured VDR mRNA levels via qRT-PCR (Figure 1B). This extended dataset validated our initial conclusion that the sphere formation frequency of calcitriol-treated GSCs negatively correlates with VDR expression (r = −0.6179). This further supports the concept of exploring calcitriol as a potential adjuvant therapy for GBM.

**Figure 1 cancers-15-05249-f001:**
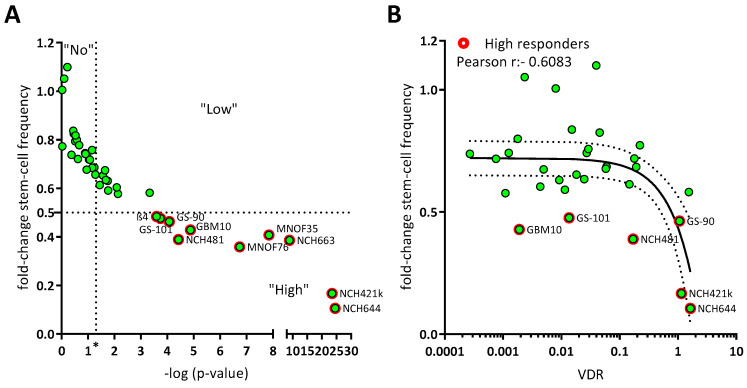
Calcitriol reduces sphere formation in a subset of GSC, which can partly be explained by VDR mRNA expression. (**A**) Point plot of the fold change of the calculated stem-cell frequency (normalized to solvent-treated GSCs) plotted against the—log *p*-value determined via Chi2-Test using ELDA-Webapp. The GSCs were summarized as non-responder (“No”), not having met the *p*-value cut-off (*p* < 0.05; vertical dotted line at “*”), low responder (“Low”) which show a significant reduction in sphere formation, but moderate effect strength (fold change > 0.5) and high responder (“High”) meeting both cut-offs (red outline). Each green point represents the mean sphere formation calculated from at least 2 experiments performed with 12 replicates and the corresponding *p*-value. (**B**) Point plot of the fold-change stem-cell frequency plotted against the VDR mRNA expression levels and Pearson correlation (black line + 95% confidence interval; dotted line). High responders are marked by red outlines.

### 3.2. Correlation of Differential Sensitivity to Calcitriol with VDR Polymorphisms

Intrigued by the fact that VDR expression levels only partially explain the sensitivity of GSCs against calcitriol, we next wondered whether analysis of VDR polymorphisms could help to elucidate this matter. Here, the four best-described VDR polymorphisms were taken into account: FokI with alleles F and f with the genotype ff, possibly indicating a higher sensitivity of cells against calcitriol, and the combined BsmI-ApaI-TaqI polymorphisms, with the “baT” genotype potentially also leading to increased sensitivity [46]. Using Restriction Fragment Length Polymorphism (RFLP) analysis, polymorphisms were analyzed on 9 high-responding as well as 8 non-responding GSC lines.

After successful amplification via specific primers (for details see materials and methods), DNA fragments were digested using the respective restriction enzyme and separated on a 2% agarose gel. Examples of RLFP patterns are shown for GBM10, MNOF35 and MNOF75, as determined via gel electrophoresis (Figure 2A). Genotypes were evaluated and are summarized in Appendix A.

Although no significant differences in response between the different groups of GSCs (by genotype) can be shown, Figure 2B shows that the portion of high responders with the more sensitive ff genotype is higher and decreases with less active VDR polymorphisms. For the least active FF genotype, the data show a strong opposite trend, with a high portion of non-responders and very few high responders. Therefore, this two-fold difference suggests that in the cell lines used for this project, the FokI polymorphism correlates with the calcitriol response. Looking at the bat genotype (Figure 2C), where baT is associated with higher and BAt with lower VDR activity, the distribution of high- and non-responding lines seems to be quite balanced with fewer high responders having the genotypes with medium-active VDR and many having higher and less active VDR genotype. The more the genotype is associated with high VDR activity, the greater the portion of high-responding GSCs are found to exhibit this genotype. The BsmI-ApaI-TaqI polymorphism, in turn, seems not to correlate, since a big portion of high-responding cell lines features the supposedly least active VDR polymorphism.

### 3.3. Comparative Proteomics of High and Non-Responders Spotlights Reduction of Stemness and Migration-Related Pathways in High-Responding GSCs

To gain systematic insights into the differences between high and non-responders under calcitriol exposure, we treated 8 high and 8 non-responders with calcitriol and performed LC-MS-based proteomics. The selected subset is displayed in Figure 3A. Overall, this 8 vs. 8 approach did not reveal a robust distinction either based on a correlation matrix (Figure 3B) or via a cluster analysis (Figure 3C). Hence, only a limited amount of differentially regulated proteins could be found. Therefore, we reduced our dataset to 5 GSCs each, resulting in a robust distinction between high and non-responders (Figure 3D) and a suitable amount of differentially regulated proteins (Figure 3E). A total of 898 and 626 significantly decreased and increased, respectively.

Using this dataset, we wanted to infer which pathways are specifically regulated by calcitriol in high-responsive GSCs. Assuming that non-responders would only show minor proteomic changes, we anticipate that the differentially regulated proteins are changes in the high-response group. First, we performed a pathway-based analysis of the significantly regulated proteins using the STRING database (https://string-db.org/; [51,52], last accessed on: 2 March 2023) (Figure 3F). With this approach, we could determine that depleted proteins after calcitriol treatment are associated with processes related to cell cycle, migration, developmental processes, and NF-κB signaling in the responders. Increased proteins in this group are associated with active translation and the ribosome, mTOR-signaling, and the unfolded protein response (UPR). Secondly, we analyzed our total dataset including the log-fold change values in order to perform enrichment analysis using STRING (Figure 3G). This analysis generally reflected the results of the restricted analysis, but revealed some additional details. Accordingly, several processes related to cell migration are depleted, including “cell adhesion molecules” and “adherens junction”, indicating an overall reduction of cell-adhesive properties. Additionally, the process “kinetochore”, a central component during cell division, was reduced, as well as “cellular response to interferon-gamma”, indicating changes relevant to immune regulation similar to NF-κB. Among the enriched processes, the overarching term can be described as translation including “ribosome”. Lastly, we also performed an enrichment analysis using the PANTHER database (http://www.pantherdb.org/; [53,54,55,56]; Figure 3H). Here, we could also determine that processes related to cell adhesion and IFN signaling are depleted. Interestingly, we also noted that processes related to developmental biology are depleted including “brain development” and “nervous system development”, reminiscent of our previous report, demonstrating reduced stemness after calcitriol treatment [11]. Among the enriched pathways, we noted translation again as a major term but also processes related to the TCA cycle, as well as cell migration. Notably, the process of “negative regulation of cell migration” is in line with our previous analysis, suggesting reduced migratory potential after calcitriol treatment.

### 3.4. Calcitriol Is Active Ex Vivo and Enhances the Effects of TMZ

Following our conclusion that calcitriol has potential as a GBM therapy, we next employed our well-established adult organotypic brain slice culture model [11,41,43] to validate our in vitro findings in a more complex tumor growth model. Previously we analyzed two of the highest responding GSC models, NCH644 and NCH421k, and determined that calcitriol effectively reduced tumor growth in both, while it also evoked combined therapeutic effects with the standard-of-care chemotherapy TMZ in NCH644 [11]. Notably, NCH644 are resistant to TMZ, due to MGMT-expression, while NCH421k are sensitive to TMZ, thereby potentially masking combinatorial effects. To address the latter issue, we reduced the TMZ concentration used in this study from 500 µM to 50 µM, which is more comparable to drug concentrations achievable in patients. From our 8 remaining high responders, we established 4 GSC cultures that could successfully be transduced and grown on OTCs, namely Beta4, GBM10, GS-101 and NCH481, and performed OTC-based tumor growth experiments (Figure 4). This approach revealed that Beta4-tumors (Figure 4A) grow very slowly and only reach 2.5 times their initial size. Treatment with calcitriol was the most effective of all and led to significantly smaller tumors 8 days after transplantation, which could further be decreased towards the end of the experiment. Beta4-tumors treated with TMZ alone and in combination with calcitriol first induced a steady-state growth until 8 days post transplantation and started to decline afterward, being significantly smaller than control tumors. No combinatorial pharmacological effects can be inferred for this line. In contrast, GBM10 tumors (Figure 4B) grow faster and reach 5 times their initial size after 15 days. These tumors are largely resistant to calcitriol with only a slight trend towards smaller growth visible. TMZ, however, blocks tumor growth within the first week and causes a decline in tumor size afterward. The combination is slightly more effective than TMZ alone, likely reflecting the minor effect calcitriol alone evokes. Hence, no combined effects can be determined in this line as well. GS-101 tumors (Figure 4C) grow continuously over the experimental time and reach 8 times their initial size with solvent treatment. Calcitriol is moderately active in these tumors and slows tumor growth significantly at 8 days post transplantation and can almost achieve a plateau in the growth curve. TMZ is similarly active, although somewhat more potent. The combination of both drugs elicits the strongest tumor growth inhibition. Accordingly, these tumors are significantly smaller compared to calcitriol single treatment. Lastly, we analyzed NCH481 GSCs (Figure 4D) and tumors derived from this line also show a continuous growth to approximately 8 times their initial size after 15 days. Both single-drug treatments appear to be equally effective in reducing tumor growth, leading to a significantly reduced tumor size 15 days post transplantation. The combination is far more effective. As such, at 8 days post transplantation, these tumors are significantly smaller compared to solvent-treated tumors, as well as both single-drug treatments. This remained true after 15 days as well, suggesting that the addition of calcitriol strongly enhances TMZ. In summary, we confirmed calcitriol single treatment to be effective in 3 out of 4 analyzed GSC cultures, as well as the two GSC lines (NCH644, NCH421k [11]) from our previous report. Additionally, we obtained data showing that the combined administration of the two drugs exerts a more potent effect than the individual treatments at least in 2–3 of the GSCs models (NCH644 [11], GS-101 and NCH481). However, this effect is clearly non-synergistic and non-additive, although it could be biologically and therapeutically relevant. No conclusion can be drawn from 2 cultures (NCH421 [11], Beta4) due to very high sensitivity to one single drug treatment.

### 3.5. Calcitriol Prevents Patient-Derived Organoid Growth

Next, we wanted to test our hypothesis on an even more translational model system and employed a set of patient-derived organoids directly established from surgical specimens. For this purpose, tumor tissue is obtained and single-cell suspensions are prepared, which can be reconstituted into organoids (Figure 5A). A representative image of an organoid is shown in Figure 5B. The quantification of viability across all nine organoids shows a slight trend for sensitivity towards calcitriol (Figure 5C). However, by analyzing each organoid culture individually, we determined that 6 PDO cultures can be classified as non-responders (Figure 5D), while 33% of these PDOs (3 of 9) are responsive to calcitriol (Figure 5E), reflecting our results from the cell-based assay as well as the OTC approach. Curiously, all responsive PDOs harbor an MGMT promoter methylation, whereas only 2 of 6 non-responding PDOs do.

## 4. Discussion

Cancer remains one of the most fatal scourges of humankind and continues to have a dismal prognosis because, among other factors, there are high rates of recurrence and devastating side effects from broadly acting chemotherapies. This is particularly true for glioblastoma, the most common malignant brain tumor. Recurrence rates are exceptionally high and novel treatment options are scarce. One approach to addressing this issue and alleviating disease burden on a population scale is to improve patient stratification for directed treatment approaches. Recently, we could show, using a restricted cell panel consisting of 10 GSCs, that 6 of those show a significant response to calcitriol, while 2 of those respond exceptionally well [11]. This result was interesting because it reflected an anecdotal clinical report from 2 decades prior showing that out of 11 patients (10 GBM, 1 anaplastic astrocytoma (AA, grade III tumor)), 3 patients (2 GBM, 1AA) responded very well to a therapy using the synthetic VitD_3_-derivative alfacalcidol leading to continuous improvement, even complete tumor regression and survival of the 2 GBM patients for more than 4 years [58]. This coherent report of our findings led us to hypothesize that this ratio might be generalizable and potentially could be employed for translational investigations. Accordingly, we expanded our cell panel with 31 additional GSC lines and could validate our initial hypothesis. Hence, approximately 50% of GSCs show a significant response to calcitriol, while 25% (10 GSCs) show a biologically meaningful response strength. Strikingly, using primary patient-derived organoids, we could independently replicate these results corroborating the notion of a VitD_3_/calcitriol-based adjuvant treatment. In fact, the application of VitD_3_/calcitriol has already been proposed in two independent position papers by Elmaci et al. and Norlin et al. a few years ago [25,26]. This proposition was mainly based on experiments using conventional cell lines and thus our initial report [11] as well as the current manuscript adds further weight to this hypothesis, particularly considering our extensive usage of stem-like spheroid lines and state-of-the-art translational models. One point that could be addressed in future studies is to determine the combined effects of calcitriol and a conventional treatment such as TMZ. While our data suggest at least an additive effect or an enhancement of TMZ in some GSC lines, it seems advisable to address this matter systematically in high- and low-responding GSCs, e.g., using the Chou–Talalay method [59] as we did previously [41], preferably further, including additional standard chemotherapeutics.

By comparing the proteomes of 8 high- and 8 non-responding GSCs treated with calcitriol, we revealed that only a small proportion of proteins is significantly regulated, partly because of a low signal-to-noise ratio, reflecting the known intertumoral heterogeneity typical for GBM. By restricting our analysis to a subset of 5 GSCs each we could instead show a clear grouping of non and high responders using a correlation analysis. This resulted in several hundred significantly regulated proteins. Assuming that non-responders do not display a considerable response to calcitriol, we concluded that the proteins and associated pathways that are changed reflect changes in high-responding GSCs. As such, we observed changes in immune system regulating processes such as NF-κB and IFN-signaling to be depleted. NF-κB signaling is frequently misregulated in cancer including GBM [60] and has been associated with the regulation of stemness, as well as immune evasion processes [61], while aberrant interferon signaling has been associated with increased proliferation and mesenchymal phenotype of GBM cells [62]. Similarly, processes related to hallmarks of GBM such as cell cycle and migration are decreased in the high responders, indicating that calcitriol treatment potentially blocks the proliferation of GSCs and might prevent tissue infiltration. Lastly, we found several processes related to developmental biology to be depleted indicating blockade of stemness in high responders. These results confirm our previous findings in a greatly extended cell panel and on the proteomic level [11]. Example proteins here include SOX9 and NES (Nestin), two widely accepted marker proteins of GSCs [3,4,63,64]. Among the enriched pathways are several associated with a stress response, such as UPR and mTOR, as well as processes related to translation. This increase in translation might reflect the activity of the VDR, which can act to regulate the transcription of target genes, leading to the observed changes in the other processes.

The clear distinction between our initial report [11] to this manuscript is, first and foremost, numbers. Hence, while our initial study indicated the general efficacy of calcitriol, we consider the current data with 40 spheroid cultures derived from several independent labs to be evidence. Secondly, we could confirm our hypothesis that calcitriol acts by blockade of stemness programs, at least in high-responding GSCs, while we could gain additional insights into further pathways that are differentially regulated after calcitriol treatment (see discussion above). Lastly, we could gain additional molecular insights into a potential biomarker for calcitriol sensitivity: VDR-polymorphism.

In the past, it has been assumed that basal expression levels of VDR are responsible for the sensitivity of cells to calcitriol. However, more recent findings indicate that instead, or additionally, VDR polymorphisms could play a role in the differential responses of tumor cells to calcitriol, as observed in human leukemia and lymphoma lines [46]. There are several polymorphisms that have been found in the coding sequence of the VDR gene, with the best-described VDR polymorphisms being defined as FokI, BsmI, ApaI and TaqI. Using RFLP analysis, these four VDR polymorphisms were analyzed in a wide panel of high-responding as well as non-responding GSC lines. FokI with alleles F and f with the genotype ff (all VDR DNA is restricted by FokI) potentially indicate higher sensitivity of cells towards calcitriol, and the BsmI-ApaI-TaqI polymorphism, with the “baT” genotype (BsmI and ApaI can digest the VDR DNA, TaqI cannot) that was also proposed to correlate with increased sensitivity [46]. Setting calcitriol responses of each GSC line in direct relation to the genotype, it could be shown that the proportion of high responders with the ff genotype, which results in a transcriptionally more active VDR variant, is higher and decreases with the less active genotype. Therefore, based on our cell panel, we conclude that the F genotype seems to positively correlate to the calcitriol response. In turn, a relationship between the genotype and sensitivity to calcitriol could not be shown concerning the baT genotype. A great portion of high-responding cell lines was found to feature the least active (BAt) genotype. These data represent an important contribution to understanding the cellular features determining calcitriol response in GBM. If VDR polymorphisms as well as mRNA expression levels [11] are detected beforehand, this combined information could be used as a biomarker to select patients that would likely profit most from adjuvant therapy using calcitriol. Based on our in vitro, ex vivo and PDO data we propose that approximately every fourth patient could benefit from a calcitriol-based adjuvant therapy. This proposition is further supported by the fact that the drug is readily available without the need for prior clinical toxicity testing, making clinical testing feasible. In a clinical setting, a possible approach could be to use the resected tumor for PDO and/or cell line generation, while concomitantly testing VDR mRNA expression levels, as well as polymorphism state and performing an in vitro interrogation for calcitriol sensitivity of the derived cultures. Patients with responsive tumor cultures could then be given high-dose calcitriol/VitD_3_ on a regular basis.

## 5. Conclusions

In conclusion, we could successfully show that calcitriol has an unmet potential as an adjuvant therapy for a subset of GBM patients and could delineate that the mechanism of action includes blockade of stemness, reduced proliferation, and potentially reduced migration/invasion. There is strong evidence that high VDR expression and the ff phenotype can serve as predictors for treatment success. Further in-depth research using even larger cell panels and/or tumor tissue should be performed to elucidate additional marker proteins and/or gene variants.

## Figures and Tables

**Figure 2 cancers-15-05249-f002:**
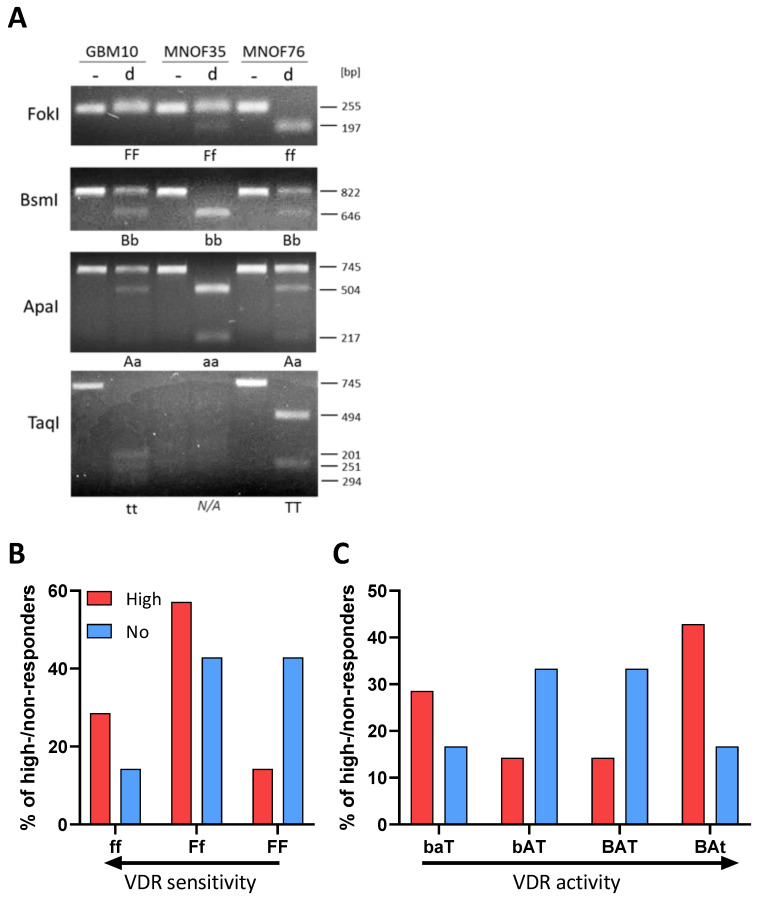
Polymorphisms of the vitamin D receptor gene. (**A**) Alleles are shown of GSCs in the gene coding for the vitamin D receptor. Shown are examples of RFLP patterns (- for undigested, d for digested) for GBM10, MNOF35 and MNOF75. Other cell lines were analyzed as well, with the patterns shown being representative. Below each gel picture, alleles are described by capital (homozygote for mutated restriction site) or small letters (homozygote for existing restriction site) or both (heterozygote for both). As found during Western blot analysis, the TaqI VDR fragment harbors another permanent TaqI restriction site that is not annotated in the database (Ensembl), which is why this fragment is restricted once or twice, resulting in two or three fragments, respectively, depending on the polymorphism (present or mutated annotated TaqI restriction site). (**B**,**C**) Distribution of high and non-responders harboring the different VDR polymorphisms. Shown are the relative fractions of high (red) and non-responders (blue) in % possessing the (**B**) FokI or (**C**) BsmI-ApaI-TaqI genotypes, respectively. See Appendix A for the original image of the agarose gels.

**Figure 3 cancers-15-05249-f003:**
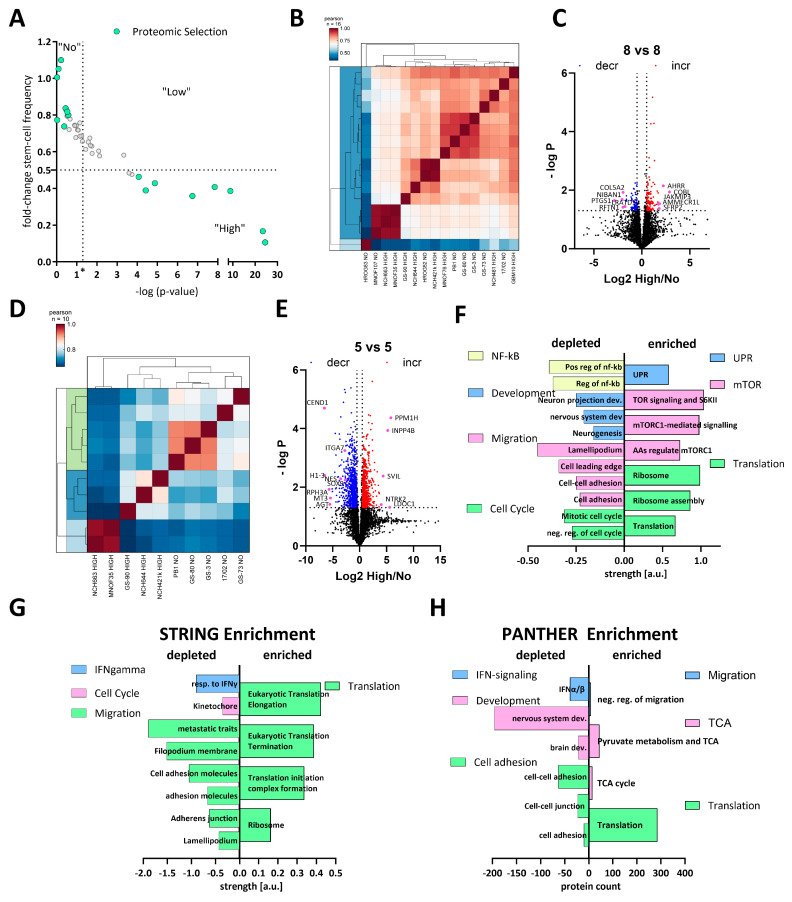
Comparative proteomics of calcitriol-treated high- and non-responding GSCs. (**A**) Point plot of the fold change in stem-cell frequency plotted against the—log *p*-value (see Figure 1 for details) with the selected GSCs marked in mint green. (**B**) Correlation plot of the entire dataset. (**C**) Volcano plot of differentially regulated proteins of the entire cell cohort (8 high and 8 non responders), showing only a few proteins consistently regulated. (**D**) Correlation plot of a subset (5 GSCs each) displaying proteomic distinction of high and non-responders. (**E**) Volcano plot of the subset of GSCs displaying 898 depleted proteins (blue dots) and 626 enriched proteins (red dots). (**F**) Pathway analysis using STRING of the significantly depleted (left side) and enriched (right side) proteins displayed in (**E**). The depleted proteins are part of the groups NF-κB (positive regulation of i-kb kinase/nf-kb signaling (GO:0043123); regulation of i-kappab kinase/nf-kappab signaling (GO:0043122)), development (neuron projection development (GO:0031175); central nervous system development (GO:0007417); neurogenesis (GO:0022008)), migration (cell adhesion (GO:0007155); cell–cell adhesion (GO:0098609); cell leading edge (GO:0031252); lamellipodium (GO:0030027)) and cell cycle (negative regulation of cell cycle (GO:0045786); mitotic cell cycle process (GO:1903047), while the enriched proteins can be grouped into UPR (unfolded protein response (HSA-381119), mTOR (TOR signaling and ribosomal protein S6 kinase II (CL:9171); mTORC1-mediated signaling (HSA-166208) amino acids regulate mTORC1, and protein kinase C terminal domain (CL:9049)) and translation (translation (GO:0006412); ribosome assembly (GO:0042255); ribosome (hsa03010)). (**G**) Enrichment analysis of the entire dataset using STRING by submitting the gene name and the corresponding log2 fold change in protein expression. Depleted pathways can be grouped under IFN-gamma (cellular response to interferon-gamma (GO:0071346)), cell cycle (kinetochore (GO:0000776)) and migration (amplification and expansion of oncogenic pathways as metastatic traits (WP3678); filopodium membrane (GO:0031527); cell adhesion molecules (hsa04514); cell–cell adhesion via plasma membrane adhesion molecules (GO:0098742); adherens junction (GO:0005912); lamellipodium (GO:0030027), while enriched pathways cluster under the term translation Eukaryotic Translation Elongation (HSA-156842); Eukaryotic Translation Termination (HSA-72764); Translation Initiation Complex Formation (HSA-72649); ribosome (GO:0005840)). (**H**) Enrichment analysis using the PANTHER web app by submitting gene name and log2 fold-change values. Processes among the depleted proteins can be grouped as IFN signaling (interferon-alpha/beta signaling (R-HSA-909733)), development (brain development (GO:0007420); nervous system development (GO:0007399)) and cell adhesion (cell adhesion mediated by integrin (GO:0033627); cell–cell junction organization (R-HSA-421270); cell–cell adhesion (GO:0098609)), while enriched proteins fall under the categories migration (negative regulation of cell migration (GO:0030336)), TCA (tricarboxylic acid cycle (GO:0006099); pyruvate metabolism and citric acid (TCA) cycle (R-HSA-71406)) and translation (translation (R-HSA-72766)).

**Figure 4 cancers-15-05249-f004:**
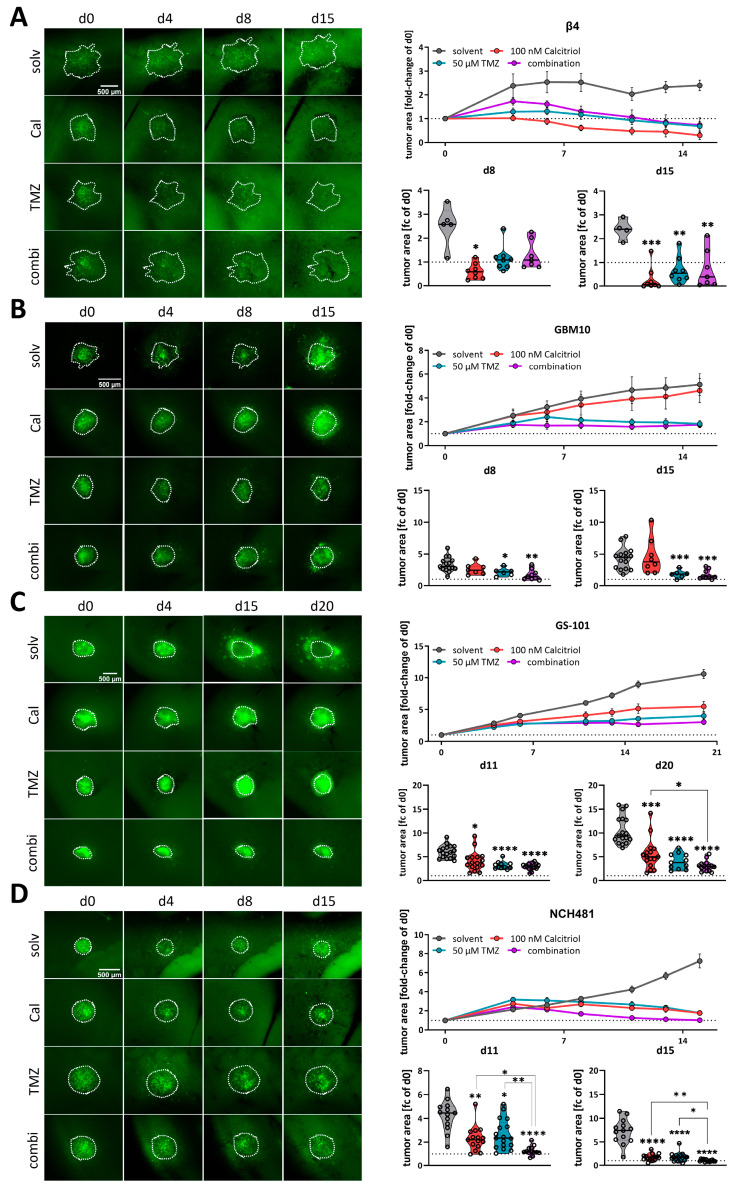
Analysis of calcitriol and combinatorial effects with TMZusing murine, adult organotypic tissue slice culture (OTC) tumor growth assay. (**A**–**D**, **left panel**) Representative microphotographs of selected time-points of tumors grown on OTCs and treated with solvent (solv), 100 nM calcitriol (Cal), 50 µM Temozolomide (TMZ) and a combination of both (combi). (**A**–**D**, **right panel**) Upper graph: tumor growth over the course of the experiment after normalizing each tumor to its size on d0. Lower graphs: violin plots of selected time-points and statistical analysis; the line within the plots represents the mean of all values. Each point represents the fold-change size of one tumor; the dashed line in one row of images indicates the size at d0 (fold change = 1). The following GSCs were analyzed: (**A**) β4, (**B**) GBM10, (**C**) GS-101 and (**D**) NCH481. * *p* < 0.05; ** *p* < 0.01; *** *p* < 0.001; **** *p* < 0.0001 against solvent treatment or as indicated; Brown-Forsythe and Welch ANOVA tests with Dunnett’s T3 multiple comparison test (GraphPad Prism 9.5).

**Figure 5 cancers-15-05249-f005:**
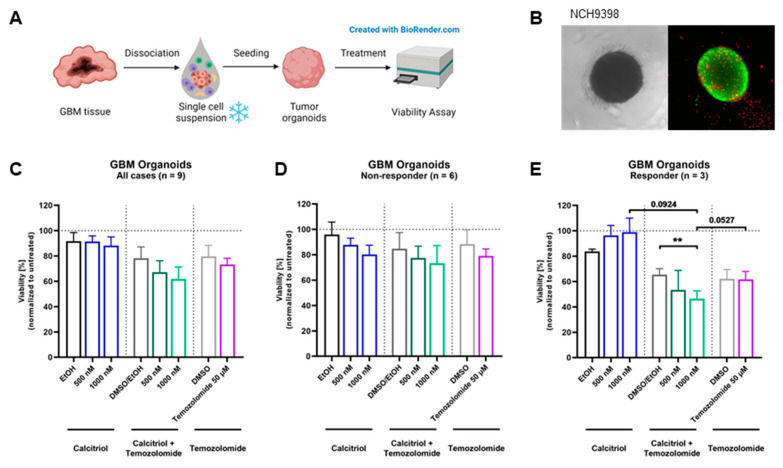
Treatment with calcitriol and Temozolomide in a patient-derived GBM organoid model: (**A**) Patient-derived GBM organoids were generated by bioprinting cells from single-cell suspensions. Four-day-old tumor organoids were treated for 5 days with calcitriol and/or Temozolomide compared to the respective controls (n = 9). On day 9, viability was analyzed. (**B**) Images of an untreated 7-day-old GBM tumor organoid in bright field and in fluorescence microscopy with a live–dead staining, depicting live cells in green and dead cells in red. (**C**–**E**) Quantification of viability analysis normalized to untreated control for GBM tumor organoids after treatment with calcitriol and/or Temozolomide compared to the respective controls (for all cases, n = 9 (**C**), non-responder cases, n = 6 (**D**), and responder cases, n = 3 (**E**). Error bars depict standard error of the mean (SEM). ** *p* < 0.01.

**Table 1 cancers-15-05249-t001:** Case overview of patient-derived GBM tumor samples for organoid model used in the presented data in Figure 1 indicating MGMT promoter methylation status and treatment response.

Case	MGMT Promoter Methylation Status	Treatment Response
NCH8295	Negative	Non-responder
NCH9398	Negative	Non-responder
NCH9403	Negative	Non-responder
NCH9514	Positive	Non-responder
NCH9538	Negative	Non-responder
NCH9608	Positive	Non-responder
NCH8138	Positive	Responder
NCH8282	Positive	Responder
NCH9659	Positive	Responder

**Table 2 cancers-15-05249-t002:** PCR parameters used for amplification of VDR polymorphism.

ApaI and TaqI			
Cycle Step	Temperature (°C)	Time (s)	Cycles
Initial Denaturation	95	300	1
Denaturation	95	30	37
Annealing	60	15	37
Extension	68	45	37
Final Extension	68	2	1
Hold	4	∞	1
**BsmI**			
Cycle step	Temperature (°C)	Time (s)	Cycles
Initial Denaturation	95	300	1
Denaturation	95	30	31
Annealing	61	15	31
Extension	68	45	31
Final Extension	68	2	1
Hold	4	∞	1
**FokI**			
Cycle step	Temperature (°C)	Time (s)	Cycles
Initial Denaturation	95	300	1
Denaturation	95	30	40
Annealing	59	30	40
Extension	68	45	40
Final Extension	68	2	1
Hold	4	∞	1

**Table 3 cancers-15-05249-t003:** Lengths of PCR amplified VDR fragments. Shown are the lengths of the gene fragments harboring the restriction sites for determination of the different polymorphisms, as well as DNA fragments resulting from restriction digestion with BsmI, ApaI, TaqI or FokI. TaqI and ApaI restriction sites are located on the same segment of the VDR gene. The ApaI/TaqI VDR fragment contains an intrinsic TaqI recognition site leading to two fragments in the absence of the TaqI polymorphism and three fragments in the presence of it.

DNA Fragment	Length(s) (bp)
BsmI VDR fragment	822
ApaI/TaqI VDR fragment	745
FokI VDR fragment	255
BsmI-digested BsmI VDR fragment	646, 176
ApaI-digested ApaI VDR fragment	504, 217
TaqI-digested ApaI/TaqI VDR fragment	494, 251 or 201, 251, 294
FokI-digested FokI VDR fragment	58, 197

## Data Availability

The mass spectrometry proteomics data have been deposited to the ProteomeXchange Consortium via the PRIDE [57] partner repository with the dataset identifier PXD041634. Additional raw data can be made available upon reasonable request.

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
