# Peer review of "Molecular Determinants of Calcitriol Signaling and Sensitivity in Glioma Stem-like Cells"

_cancers, 2023, doi:10.3390/cancers15215249_

Round 1
Reviewer 1 Report
Comments and Suggestions for Authors
There is overlap with the concepts in this manuscript with those published recently in ref 15 by the same authors. This significantly reduces the impact of the publication.
Focusing on Fig 4. highlights the fundamental issue. The authors claim that Figures 4C and 4D show synergy with the combination of calcitriol and TMZ for the 2 cell lines. There is no synergy! Indeed, the date is less than additive. This claim is simply not justified at all. If the authors believe it is synergistic, please provide the data (mathematics) proving this statement. Furthermore the date in Figure 5E is also additive at best (probably less than additive), highlighting the lack of synergy.
The labels for days of treatment on Fig 4D are completely wrong, making it impossible to assess.
The day 15 and 20 solvent samples do not even circle the cells. How do the authors calculate area if the circle does not include all the cells.
The concerns with this figure alone highlight the shortcomings of this manuscript.
Labelling on Figure 3F-H is messy.
Comments on the Quality of English LanguageThe English only need minor editing.
Author Response
There is overlap with the concepts in this manuscript with those published recently in ref 15 by the same authors. This significantly reduces the impact of the publication.
Answer: We agree that there is some overlap in the overall concept of the previous and current studies. We would like to point out that there are important and completely novel aspects in the submitted work (role of gene polymorphisms, proteomic changes in high responders vs low responders), and the extent of overlap between the two studies is reasonable in our opinion. The present work is the logical continuation of the previous study and offers additional new insights, particularly concerning more confidence in the cell culture data by a 4-fold increase in GSC lines used and by adding several advanced models to enhance the translational impact of our work (OTCs and GBM organoids). Focusing on Fig 4. highlights the fundamental issue. The authors claim that Figures 4C and 4D show synergy with the combination of calcitriol and TMZ for the 2 cell lines. There is no synergy! Indeed, the date is less than additive. This claim is simply not justified at all. If the authors believe it is synergistic, please provide the data (mathematics) proving this statement. Furthermore the date in Figure 5E is also additive at best (probably less than additive), highlighting the lack of synergy.
Answer: We would like to thank the author for this remark. While we agree that a mathematical approach surely can give valuable insights, it usually requires several concentrations of each drug as well as their combination, even when done using pragmatic approaches like the Chou-Talalay-Method. Hence, it is our believe that these approaches are realistic/have value for purely cell culture-based pharmacological assays with at least a moderate throughput (i.e. 96-well plates) to guide further experiments. For a more complex model such as OTCs these approaches lead to an unnecessary high amount of samples needed to gain any reliable data output, and in this case also requires the sacrifice of more animals. For these reasons we went for the less stringent approach in defining synergism as a significant improvement compared to single drug treatment, which we clearly stated in the results part. To address the issue, we have now discussed the implementation of a cell-culture-based synergism calculation and changed the wording regarding synergism.
The labels for days of treatment on Fig 4D are completely wrong, making it impossible to assess.
Answer: Thanks a lot for pointing this out. Something went wrong during the export from GraphPad-Prism and we have now fixed this issue.
The day 15 and 20 solvent samples do not even circle the cells. How do the authors calculate area if the circle does not include all the cells.
Answer: There seems to be a misunderstanding. The circle always (!) refers to the tumor size at d0 in order to highlight the change in size over time. We have now clarified this in the figure legends.
The concerns with this figure alone highlight the shortcomings of this manuscript.
Labelling on Figure 3F-H is messy.
Answer: We are very sorry that the reviewer feels like this. However, we have used this way of presenting before, because it is, in our opinion, the most accurate way. Nonetheless, we agree that the figure lacked clarity and have shortened the pathway names and now provide the full pathway name in the figure legends.
Reviewer 2 Report
Comments and Suggestions for Authors
Title: Molecular Determinants of Calcitriol Signaling and Sensitivity 2 in Glioma Stem-like Cells
Authors: Sarah Rehbein et al.
Journal ID: Cancers 2587110
The antiproliferative effects of hormonally active form of vitamin D3 have been demonstrated in various cancer types, as determined by preclinical trials. In this study, the authors have made some important contributions to the understanding of cellular heterogeneity that determines the response to calcitriol with elegant in vitro, ex vivo and functional studies.
This is a hypothesis-driven, well executed and nicely written manuscript and I have no hesitation to recommend the Section Editors to accept it as is.
Author Response
We would like to thank the reviewer for the appraisal of our work and appreciate the time to review our manuscript.
Reviewer 3 Report
Comments and Suggestions for Authors
The manuscript is well-written and the assays are well designed. The authors have previously published an article that provides the groundwork for this manuscript.
The researchers conducted experiments to discover a new treatment for the most common primary brain cancer, which is one of the most severe types of cancer for patients. The experiments were comprehensive and involved several cell lines to confirm the findings.
I am willing to accept the current manuscript without any additional experiments. However, I have a question regarding the polymorphism analysis. Why were there 9 cell lines used in the experiment, while only 8 high responders were used in the proteomics assay? Were the same 8 cell lines used in both experiments and analysis? I suggest adding Figure 3A to the polymorphism assay or referring to it if the selected cells are the same. It would help to clarify the selected cells.
During the STRING analysis, the signatures might come from different databases, such as GO terms, published articles, etc. I suggest adding the source of the signatures used in Figure 3 and the related results.
Did the researchers test different concentrations of treatment alone and in combination? The results are intriguing, as they suggest that calcitriol may be a promising treatment for glioblastoma, either alone or in combination with TMZ. However, have the authors tested the toxicity of the treatment to normal cells, such as neurons or astrocytes? It would be interesting to test the treatment in in vivo assays, in normal cell culture, or using the new Glioblastoma Brain Organoid Co-Cultures model.
Author Response
The manuscript is well-written and the assays are well designed. The authors have previously published an article that provides the groundwork for this manuscript.
The researchers conducted experiments to discover a new treatment for the most common primary brain cancer, which is one of the most severe types of cancer for patients. The experiments were comprehensive and involved several cell lines to confirm the findings.
Answer: We would like to thank the review for the appraisal of our work. We appreciate the time spent on reviewing our manuscript
I am willing to accept the current manuscript without any additional experiments. However, I have a question regarding the polymorphism analysis. Why were there 9 cell lines used in the experiment, while only 8 high responders were used in the proteomics assay? Were the same 8 cell lines used in both experiments and analysis? I suggest adding Figure 3A to the polymorphism assay or referring to it if the selected cells are the same. It would help to clarify the selected cells.
Answer: That is an excellent question. This number is mainly caused by technical reasons concerning the proteomic approach. We labeled the peptides using TMT16pro, meaning that we can pool 16 samples after labeling, before proceeding the fractionation. For this reason we were restricted to 8 instead of 9 samples per group.
During the STRING analysis, the signatures might come from different databases, such as GO terms, published articles, etc. I suggest adding the source of the signatures used in Figure 3 and the related results.
Answer: Thank you for this suggestion. We have now added the source of the signatures in the result sections.
Did the researchers test different concentrations of treatment alone and in combination? The results are intriguing, as they suggest that calcitriol may be a promising treatment for glioblastoma, either alone or in combination with TMZ. However, have the authors tested the toxicity of the treatment to normal cells, such as neurons or astrocytes? It would be interesting to test the treatment in in vivo assays, in normal cell culture, or using the new Glioblastoma Brain Organoid Co-Cultures model.
Answer: Both points are valid. In our previous work we used multiple concentrations of calcitriol single drug treatment in cell cultures and could show a dose-dependency. As for the combination treatment: No, we have not done this so far, because we wanted to focus on more complex models and global approaches for this manuscript. Regarding the non-tumor cells: We have not explicitly tested other cell types in vitro, however we have performed several experiments using brain slices and never observed signs of toxicity. In addition, VitD3 and calcitriol have a favorable safety profile and are used in research and clinical practice for several decades now with the most frequent toxicity is hypercalcemia, which can be easily monitored. Nonetheless, we agree that a confirmatory set of experiments should be included in future experiments to determine the effects of glioma-associated brain cells..
Reviewer 4 Report
Comments and Suggestions for Authors
The manuscript titled "Molecular Determinants of Calcitriol Signaling and Sensitivity in Glioma Stem-like Cells" by Rehbein et al. address a critical aspect of glioblastoma treatment, focusing on the potential role of Calcitriol in reducing stemness, tumor growth, and possibly migration/invasion of glioma stem-like cells. The manuscript is well-crafted, and the conclusions are supported by clearly presented results, which offer an encouraging direction for further research and potential clinical applications.
Major Strengths:
1.Methodological Excellence: The methodology, including the limiting dilution assay, lentiviral transduction, adult organotypic brain slice cultures, ex vivo tumor growth assay, Restriction Fragment Length Polymorphism analysis, and mass spectrometry, represents state-of-the-art techniques that are apt for this kind of study. Their comprehensive approach provides robustness to the study.
2. Results and Conclusions Alignment: The authors have presented compelling results such as:
The association of genotype with high VDR activity and its correlation with high-responding GSCs.
Proteomics analysis indicating the enriched pathways that support the negative regulation of cell migration post-calcitriol treatment.
Evidence suggesting that Calcitriol synergizes with TMZ, inhibits patient-derived organoid growth, and has potential as an adjuvant therapy for a subset of GBM patients.
3. Quality of Writing: Overall, the manuscript is well-written and organized, which facilitates understanding and appreciating the significance of the work.
Suggestions for Improvement:
1. Introduction Expansion: It would benefit the readers if the introduction could be expanded to provide a more comprehensive context. A broader overview of glioblastoma, its challenges, and the need for alternative treatments like Calcitriol would set a firmer foundation for the study.
2. Paragraphing in Introduction: I suggest breaking the first paragraph into two smaller, more digestible sections. The first should revolve around glioblastoma, its prevalence, and the associated challenges. The subsequent paragraph can focus on Calcitriol, its potential role, and previous findings related to it.
In conclusion, I believe that this manuscript offers valuable insights into glioblastoma treatment with Calcitriol and holds promise for further research and clinical applications. I recommend the manuscript for publication in Cancers journal, pending the suggested minor revisions.
Author Response
We would to thank the reviewer for the appraisal of our manuscript and we appreciate the time to review our work. We have now addressed to suggestions of the reviewer and expanded the introduction and improved the structure of the manuscript.
Round 2
Reviewer 1 Report
Comments and Suggestions for Authors
The overlap of the basic finding from your previous manuscript is large. Your claims about suggestive synergy are simple wrong. There is no possibility of synergy. Indeed much of the data is less than additive!
Comments on the Quality of English LanguageThe English only needs minor corrections.
Author Response
Wording has been further improved.